# Monitoring Corrosion Processes via Visible Fiber-Optic Evanescent Wave Sensor

**Dervis Türkmen [1], Achim Krug [2] and Boris Mizaikoff [1,*]**

[1]  Institute of Analytical and Bioanalytical Chemistry, Ulm University, Albert-Einstein-Allee 11, 89081 Ulm, Germany; dervis.tuerkmen@uni-ulm.de

[2]  Wertec GmbH, Oberlaaer Straße 232, 1100 Wien, Austria; achim.krug@wertec-gmbh.at

*  Correspondence: boris.mizaikoff@uni-ulm.de; Tel.: +49-731-50-22750

**Abstract:** Ferrous objects, especially those that are additionally exposed to harsh environments, e.g., high humidity, have the common problem of suffering aggressive corrosion processes. Without a precaution, this leads in many cases to a limited functionality followed by treatment steps, and expensive repairing costs, as well as to defects/uselessness and even to safety problems, e.g., bridge-collapsing, escaping gas and liquids from pipelines, or leaking oil tankers destroying the ecological system. Conventionally applied methods are confined to laboratory use due to bulky instruments, and are, therefore, time-intensive and may cause erroneous results. Therefore, a sensor based on fiber-optic evanescent wave spectroscopy (FEWS) working in the visible spectral range was developed. The sensor system is comprised of an uncoated single crystal sapphire fiber as a waveguide operating in the visible spectral range in combination with a laser diode as a light source at a specific wavelength and a photodiode for signal detection. Within this study, the potential of the developed sensor was investigated. The corrosion process was simulated by implementing a corroded iron bar inside the measuring cell. When corrosion starts, iron ions are released leading to a complexation reaction with the dye. The results showed the potential use of the developed sensor system enabling implementation for online and on-site detection and monitoring of components susceptible to corrosion.

**Keywords:** fiber-optic evanescent wave sensor; visible attenuated total reflection; sapphire fiber; iron-sensitive dyes; corrosion detection; corrosion monitoring

## 1. Introduction

For the production of iron, its important materials and goods, an annual large amount of iron ore is mined. Every year 1670 million tons of iron are processed worldwide with an annual increase due to its use in a wide variety of sections, e.g., automotive industry, track for railways, shipyards, bridges, as well as in reinforced concrete systems as implemented in buildings and many others [1]. In addition to the advantageous properties of iron, such as high tensile strength, high load-bearing capacity, and the high variety of applications, iron has the common problem of corrosion [2,3]. This electrochemical reaction can cause a large mass loss. In combination with mechanical stress, corrosion may lead to unwanted cracking or formation of holes up to limited functionality followed by expensive repairing costs, as well as defects/uselessness and even safety problems, e.g., bridge-collapsing, escaping gas and liquids from pipelines, or leaking oil tankers destroying the ecological system [4–8].

The formation of corrosion-based rust is illustrated in Figure 1. A water droplet (blue) surrounded by air is mapped on top of an iron surface (grey). First, the rusting process begins with the oxidation, which can be described as the diffusion of positively-charged iron ions into the aqueous solution while electrons remain in the metal (A). Due to the presence of oxygen, the electrons are transported to the

oxygen-rich area (B). Thereby, a galvanic cell is formed with an anodic area (a), a cathodic area (b), and water as electrolyte. The next step is the reduction of oxygen to hydroxide ions (C). The new formed hydroxide ions interact with the existing iron ions to form iron(II) hydroxide (D). Due to further redox reactions, a hardly soluble rust complex is formed (E). The equivalent equations of reaction are given in Equations (1)–(7). The formula of the final rust complex is given in Equation (8).

$$\text{Fe} \rightarrow \text{Fe}^{2+} + 2\,\text{e}^- \tag{1}$$

$$0.5\,\text{O}_2 + \text{H}_2\text{O} + 2\,\text{e}^- \rightarrow 2\,\text{OH}^- \tag{2}$$

$$\text{Fe}^{2+} + 2\,\text{OH}^- \rightarrow \text{Fe(OH)}_2 \tag{3}$$

$$\text{Fe(OH)}_2 \rightarrow \text{FeO} + \text{H}_2\text{O} \tag{4}$$

$$2\,\text{Fe(OH)}_2 + 0.5\,\text{O}_2 + \text{H}_2\text{O} \rightarrow 2\,\text{Fe(OH)}_3 \tag{5}$$

$$\text{Fe(OH)}_3 \rightarrow \text{FeO(OH)} \cdot \text{H}_2\text{O} \tag{6}$$

$$2\,\text{FeO(OH)} \rightarrow \text{Fe}_2\text{O}_3 + \text{H}_2\text{O} \tag{7}$$

$$x\,\text{FeO} \cdot y\,\text{Fe}_2\text{O}_3 \cdot z\,\text{H}_2\text{O} \tag{8}$$

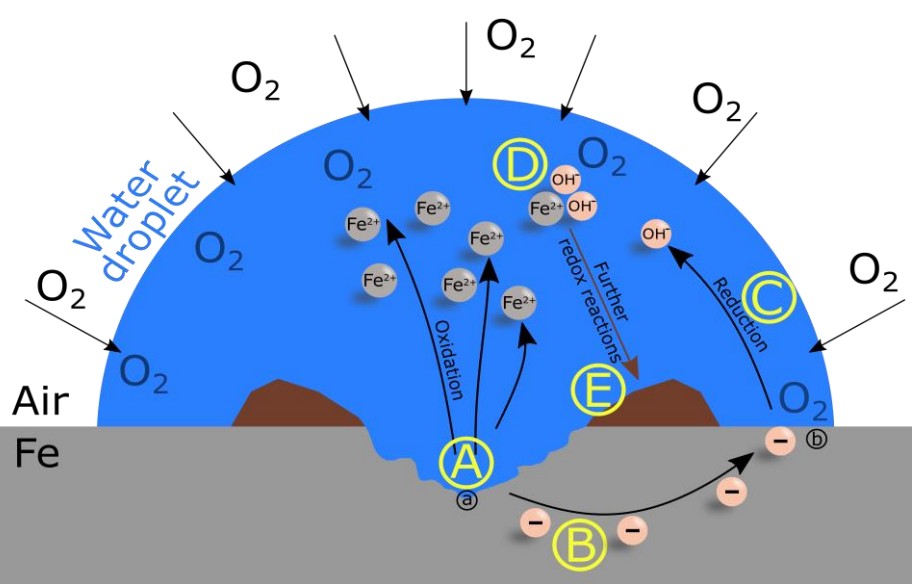

**Figure 1.** Formation of rust (brown) on an iron surface (grey) due to the presence of water (blue). Iron ions diffuse into the water droplet and electrons remain in the metal (A) that are transported to the edge of the water droplet/oxygen rich area (B). Afterwards, hydroxide ions are formed due to the reduction of oxygen (C) and are distributed in the water droplet. The chemical reaction of iron ions and hydroxide ions lead to iron-hydroxide molecules (D). A hardly soluble rust complex is formed subsequent to further redox reactions (E). The whole system resembles a galvanic cell with an anodic area (a), a cathodic area (b), and an electrolyte which is provided by the presence of water.

To prevent safety risks and following damages caused by corrosion, it is necessary to recognize rust immediately from the beginning. Previous investigations show the use of electrochemical impedance spectroscopy (EIS) to analyze corrosion processes [9–11]. Therefore, a three-electrode cell with a working electrode, counter electrode, and a reference electrode is used to perform a measurement by applying a certain voltage and current to a metal sheet placed in the measuring cell filled with an aqueous solution as electrolyte. EIS is often limited to laboratory use only due to its sensitivity to vibrations. In addition,

this method is very time-consuming due to the recording of a full impedance spectrum and its various evaluation methods and requires a good knowledge in electrical engineering [9,11,12].

Alternatively, using the Mössbauer spectroscopy, Graham and Cohen [13] analyzed corrosion products using X-ray diffraction. Therefore, low energetic $\gamma$-rays are combined using the Doppler effect to investigate solid iron products such as rust. Drawbacks of this methods are complex equipment, high costs, and the limitation of laboratory use only. Furthermore, investigations of liquid samples provide inaccurate results compared to solid samples.

Iron ions are mainly detected via complexation reactions with iron-sensitive dyes causing a visible color change. Therefore, iron ions are detectable via UV–VIS spectroscopy. However, UV–VIS spectrometer are restricted to laboratory use due to bulky instruments. Furthermore, the samples have to be diluted which is time-consuming and may cause errors. Additionally, measurements are performed in transmission mode, i.e., light has to be guided through the sample with pathlengths in the millimeter range. Therefore, conventional UV–VIS spectroscopy is not applicable for continuous monitoring directly on-site, e.g., detection of components prone to corrosion. Attenuated total reflection (ATR) has the advantage of direct measuring with optical pathlengths in the micrometer range. Conventionally, ATR is applied for analysis of molecular vibrations in the mid-infrared. With decreasing wavelengths, molecular vibrations decrease. However, materials, i.e., dyes and chromophores, owe their high absorbances to electronic transition. Previous studies have already demonstrated the feasibility of UV–VIS spectroscopy in combination with ATR for high-absorbing samples [14,15]. Measurements can be performed directly, and sample preparation is not necessary with possible application in process and quality control [14].

In this study, a sensor based on visible-ATR, i.e., FEWS, was developed. FEWS is based on a variety of advantages, such as nondestructive monitoring, real-time sampling, flexibility, extended reach, in situ measurements, and small size [4,16–18]. Therefore, FEWS enables implementation directly on-site for monitoring and surveillance of corrosion sensitive materials. In Figure 2, the principle of ATR is illustrated with a sapphire fiber as an optical waveguide. For total internal reflection to occur, the angle of the incident beam $\theta_i$ must be larger than the critical angle $\theta_C$. According to Snell's law (Equation 9) the refractive index of the transducer $n_1$ must be larger than the refractive index of the adjacent medium $n_2$ [19,20].

$$\theta_C = \arcsin\frac{n_2}{n_1} \tag{9}$$

The emerging evanescent field is caused by the interference of the incident and the reflected beam. This evanescent wave penetrates into the sample with a certain length termed as penetration depth ($d_p$) given in Equation (10) [19]:

$$d_p = \frac{\lambda}{2\pi \sqrt{n_1^2 \sin^2 \theta_i - n_2^2}} \tag{10}$$

Due to the dependence on distance $x$, the intensity $E$ can be described as:

$$E(x) = E_0 \exp\left(-\frac{x}{d_p}\right) \tag{11}$$

Fiber-optic materials with a transmission range in the visible spectral range include heavy metal fluorides (glass fiber), sapphire (crystalline fiber), and silica fibers. The attenuation losses of those fibers are relatively high with 0.08 dB/m (2.94 µm) for heavy metal fluorides, 0.1–0.4 dB/m (2.5–0.5 µm) for sapphire fibers, and 0.012 dB/m (0.82 µm) for silica fibers compared to telecommunication fibers (0.0002 dB/m at 1.5 µm) [21–23]. However, telecommunication fibers are conventionally used over several tens of kilometers, whereas fiber lengths within this sensor application are shorter. Sapphire fibers offer a high refractive index (approx. 1.702–1.834 at a wavelength of 3.303–0.265 µm) having the ability of measuring fluids, gels, and other components with high refractive index. Since the gel, which is applied around components for inhibiting corrosion, has a relatively high refractive index

of approx. 1.5, a sapphire fiber was used as optical waveguide within this study. Standard silica or fluoride fibers offer high transmission in the visible spectral range but are not applicable due to their low refractive indices [14]. Sapphire-combined FEWS sensors are increasingly becoming more important due to their chemical inertness, biocompatibility, extreme mechanical and thermal stability, and non-toxicity. Therefore, sapphire fibers can be used in a variety of applications especially in continuous operation, such as corrosion monitoring [22,24]. In addition to the inhibition function of the gel, the fiber is simultaneously protected from contamination, which has to be avoided based on the small penetration depth (hundreds of nm) of the visible light compared to the penetration depth in the near-IR with several μm [14].

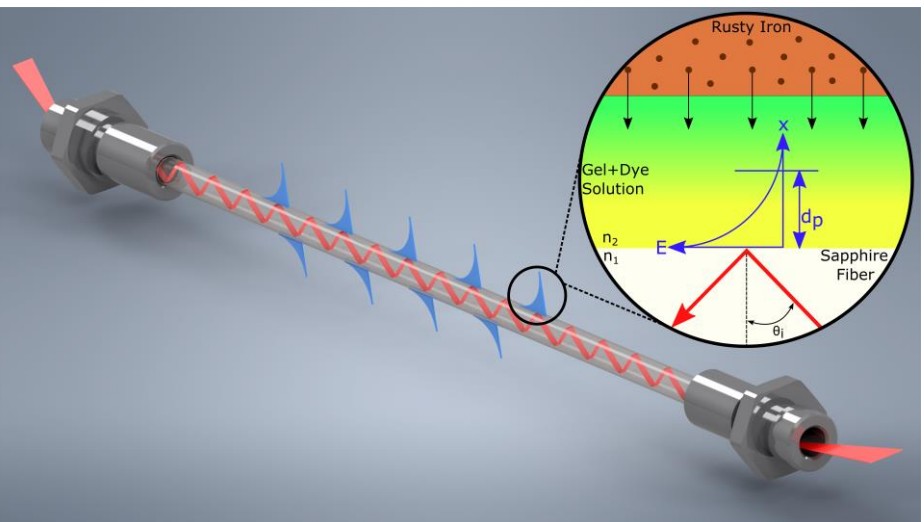

**Figure 2.** Attenuated total reflection (ATR) of an incoming beam (red) inside a sapphire fiber. Incident and reflected beam at the interface to the surrounding sample (gel/dye solution), interfere and form an evanescent wave (blue) that penetrates into the adjacent medium with a certain penetration depth ($d_p$). Thereby, spectral change of the sample can be monitored which contains an iron-sensitive dye triggered by the iron ions diffusing from a rusty iron bar (brown). The refractive indices of the fiber and sample are $n_1$ and $n_2$, respectively. $\theta_i$ is the angle of incident beam, and $E$ is the intensity of the emerging evanescent wave regarding the distance $x$.

As already mentioned, iron ions can be detected via complexation reaction with dye molecules via UV–VIS spectroscopy. The ideal dye requires a high sensitivity to iron ions, i.e., with a minimal amount of dye, even small amounts of iron ions are detectable. The dye should be of low-cost, non-toxic, and stable over a long period of time to changing weather conditions (i.e., varying temperature and humidity). Previous investigations show successful application of DHP, DHB, *N*-(rhodamine-6G)lactam-ethylenediamine, 8-hydroxyquinolin, and others in the area of iron detection [11,25,26].

The present study describes the development of a FEWS sensor operating in the visible spectral range. A sapphire fiber without cladding and jacket was used as transducer with a laser diode as the light source, and a photodiode as the detector. Various dyes capable of reacting with iron ions initiated by the presence of iron ions from a rusty iron bar were investigated at three different wavelengths with the developed sensing system showing detection limits comparable to conventional UV–VIS spectroscopy. The developed sensor has the potential to be deployed directly in-field for applications wherever corrosion may take place or lead to a safety problem.

## 2. Materials and Methods

### 2.1. Materials

DHP (95%), TPTZ (99%), DHB (99%), DG (99%), GA (98%), PFC-II (99%), PFC-III (99%), OG (99%), Tiron (97%), diethyl ether (99.7%), 2-propanol (99.9%), and ethanol (99.8%) were obtained from Sigma-Aldrich Chemie GmbH (Munich, Germany). Gel-04 was provided from Wertec GmbH and purchased from Unigel LTD (Eastbourne, United Kingdom). The composition of the gel used as filling material acting as corrosion inhibitor for materials susceptible to corrosion is found in the literature [27]. Fe(III) chloride (98%) was provided by Merck KGaA (Darmstadt, Germany).

### 2.2. Preparation of Dyes and Corrosion Samples

The respective dye was dissolved in 2 mL diethyl ether. After adding 10 g of Gel-04, the solution was stirred for two hours until a homogeneous mass was obtained. The viscous solution was hermetically stored for two days. Afterwards, the gel/dye mixture was dried for one day at 40 °C ensuring complete evaporation of diethyl ether.

A blank iron bar (5.8 mm in diameter) was cut in 22 mm pieces. The pieces were wetted with water and stored outside for three months to induce the corrosion process. The initial and corroded iron bars are shown in Figure 3 left and right, respectively.

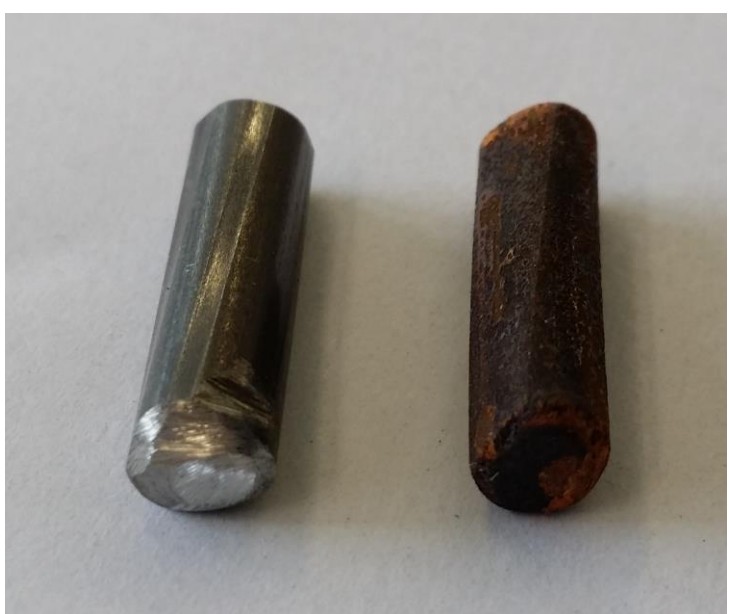

**Figure 3.** (Left) Blank uncorroded iron bar. (Right) Iron bar stored for three months outside with changing temperature and humidity initiating the corrosion process.

### 2.3. UV–VIS

UV-VIS spectra of the gel/dye mixtures were recorded on a Jasco V-670 spectrometer (Jasco, Pfungstadt, Germany) in the spectral region ranging from 250 nm to 900 nm. For each spectrum, three-hundred scans were averaged using a spectral resolution of 2 cm$^{-1}$ (wavelength of 0.5 cm). Spectra with iron ions (Fe$^{2+}$ and Fe$^{3+}$) were obtained by adding 10 μL of 10.3 mg/mL iron(II) chloride or iron(III) chloride in diethyl ether.

### 2.4. Sapphire Fiber-Optic Corrosion Sensor

A bare, uncoated sapphire fiber (250 μm in diameter, 10 cm in length), provided by Laser Components GmbH, Olching, Germany, was cleaned with 2-propanol and subsequently with diethyl

ether before installing the fiber-optic waveguide in the 3D printed polylactide (PLA) measuring cell. The corroded iron bar was mounted inside the measuring cell with a distance of 1 mm to the fiber. Finally, the prepared gel/dye mixture was added to the measurement cell. A schematic CAD drawing of the entire measuring cell is shown in Figure 4.

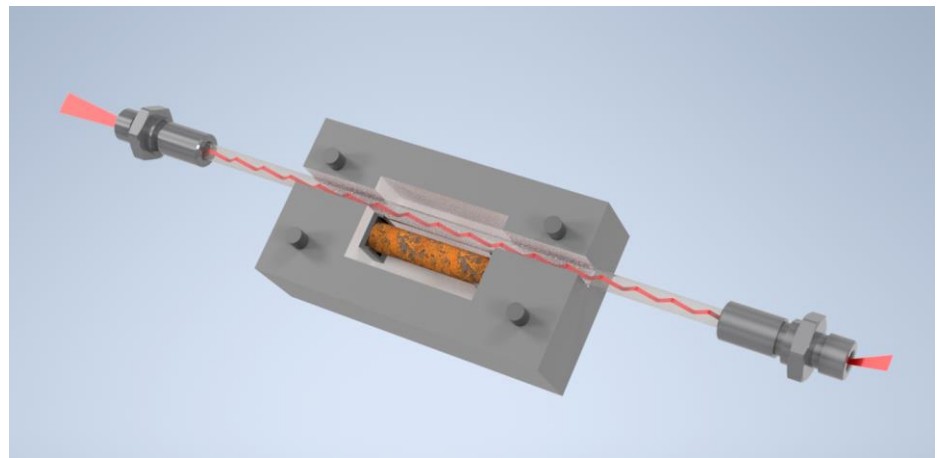

**Figure 4.** Schematic CAD drawing of the measuring cell which was 3D printed. The unclad sapphire fiber is located inside next to the component which could corrode over time. This component was simulated by placing an already-corroded iron bar inside the measuring cell.

As a light source, three pigtailed laser diodes (520 nm, 635 nm, and 705 nm) (Thorlabs GmbH, München, Germany) were used. Depending on the desired wavelength, the respective laser was mounted into the pigtailed laser diode mount. The laser diode was coupled to a laser temperature controller and a laser diode controller unit to adjust correct settings (Thorlabs GmbH, München, Germany). To obtain a stabilized, optimum fiber output power, the operating current $P_{op}$ was adjusted to a certain value as given in Table 1. A photodiode was used as the detector. Light from the laser diode was easily coupled into and out of the sapphire fiber to the photodiode detector via silica fibers with conventional SMA fiber connectors on both ends. Therefore, alignment of the sensor components is not necessary. Specifications of the utilized multimode silica fibers including the core and cladding size, the numerical aperture (NA) and the maximum attenuation at a wavelength of 808 nm are summarized in Table 2. Communication from the photodiode to the computer was provided via a USB power meter. Measurements were performed using the software APT$^{\text{TM}}$ from Thorlabs GmbH. A schematic of the entire sensor system is shown in Figure 5.

**Table 1.** Adjusted operating current and fiber output power of the laser diodes with the wavelength of 520 nm, 635 nm, and 705 nm.

| Wavelength | Operating Current | Output Power |
|---|---|---|
| 520 nm | 93.1 mA | 15.0 mW |
| 635 nm | 75.4 mA | 8.0 mW |
| 705 nm | 66.9 mA | 15.0 mW |

**Table 2.** Summary of specifications of multimode silica fibers utilized for light guidance in and out of the sapphire fiber [28].

| Core Diameter | Cladding Diameter | NA | Maximum Attenuation at 808 nm |
|---|---|---|---|
| 400 μm | 425 μm | 0.39 | 10 db/km |

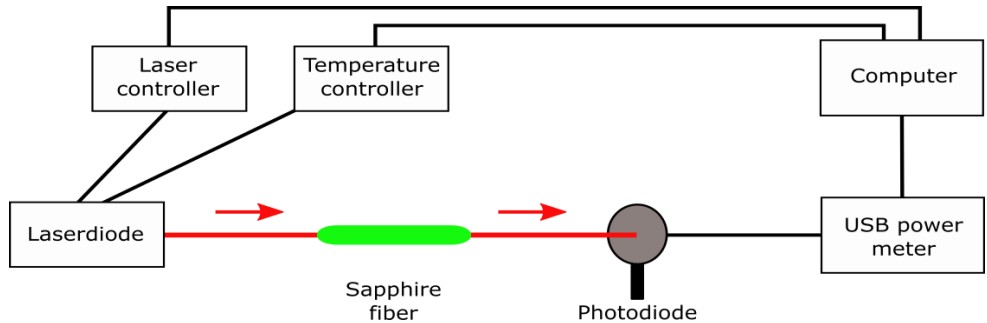

**Figure 5.** Schematic of the fiber-based corrosion sensor comprising a laser diode as light source with laser and temperature controller, sapphire fiber as waveguide, and a photodiode as detector. The sapphire fiber is coupled with the laser diode and the photodiode via a silica fiber for light propagation. The USB power meter provides communication between the photodiode and the computer.

One measurement includes 1000 single measurement points, i.e., scans. One scan is equal to 352 ms. The gel/dye mixture was used as background spectrum. The addition of iron ions, i.e., iron ions from the corroded iron bar, corresponds to the sample spectrum. The absorbance values $A$ were calculated according to Beer–Lambert law: $A = -\log \frac{I}{I_0}$, with the intensity of the sample $I$ and the initial intensity of the background $I_0$. All measurements were performed at room temperature (22 °C). The data was evaluated using Essential FTIR $^®$ Spectroscopy software.

## 3. Results

### 3.1. UV–VIS

Nine different dyes were tested for complexation reaction using UV–VIS spectroscopy, in order to detect corrosion, and therefore, iron ions. These dyes were selected based on their use as iron detection agent. If an uncorroded iron bar is exposed to a gel/dye mixture, complexation do not occur since there are no free iron ions, hence, a color change is not observable as shown in Figure 6a exemplarily for TPTZ as dye. If corrosion takes place, iron ions are released leading to a reaction with the dye molecules and a color change, in the case of TPTZ from colorless to violet (Figure 6b).

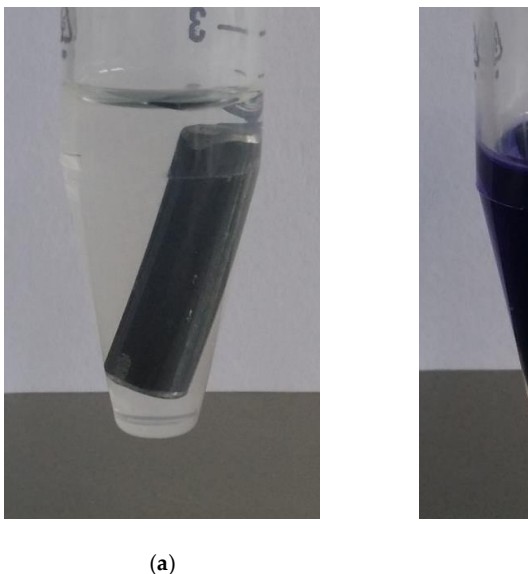
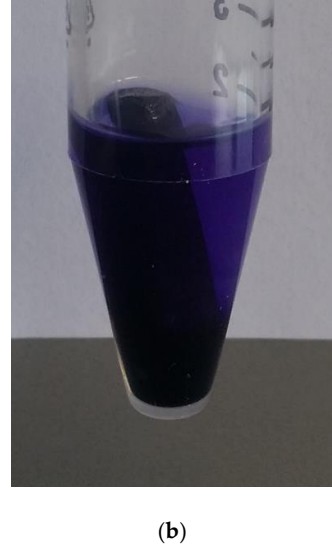

(**a**)　　　　　　　　　　　　(**b**)

**Figure 6.** A non-corroded (**a**) and corroded (**b**) iron bar was exposed to a gel/dye mixture. TPTZ was used as the dye. Due to release of iron ions based on the corrosion process, a complexation reaction occurred revealing a direct change in color.

The amount of the respective dye was very high (200 mg) to ensure a change in UV–VIS spectrum when iron ions are present. The black line in Figure 7 shows the UV–VIS absorption spectra of (a) DHP, (b) TPTZ, (c) DHB, (d) DG, (e) GA, (f) PFC-II, (g) PFC-III, (h) OG, and (i) Tiron. In order to observe a change, an excess of iron ions (10 μL of $Fe^{2+}$ and $Fe^{3+}$, both in a concentration of 10.3 mg/mL) was added. The UV–VIS spectra changed due to the complexation reaction after adding $Fe^{2+}$ or $Fe^{3+}$ ions marked as a red or blue line, respectively. Strong differences before and after adding iron ions can be observed in the spectral region from 500 nm to 800 nm. Almost all dyes respond to both, $Fe^{2+}$ and $Fe^{3+}$ ions. This is of great importance, since corrosion processes leads to the presence of both ions. Tiron only shows a small increase in absorbance with $Fe^{2+}$ ions. Furthermore, PFC-III undergoes no reaction with $Fe^{3+}$ ions, i.e., no increase in UV–VIS absorption spectrum, therefore, the UV–VIS spectrum after addition of $Fe^{3+}$ ions is similar to the pure dye (black spectrum).

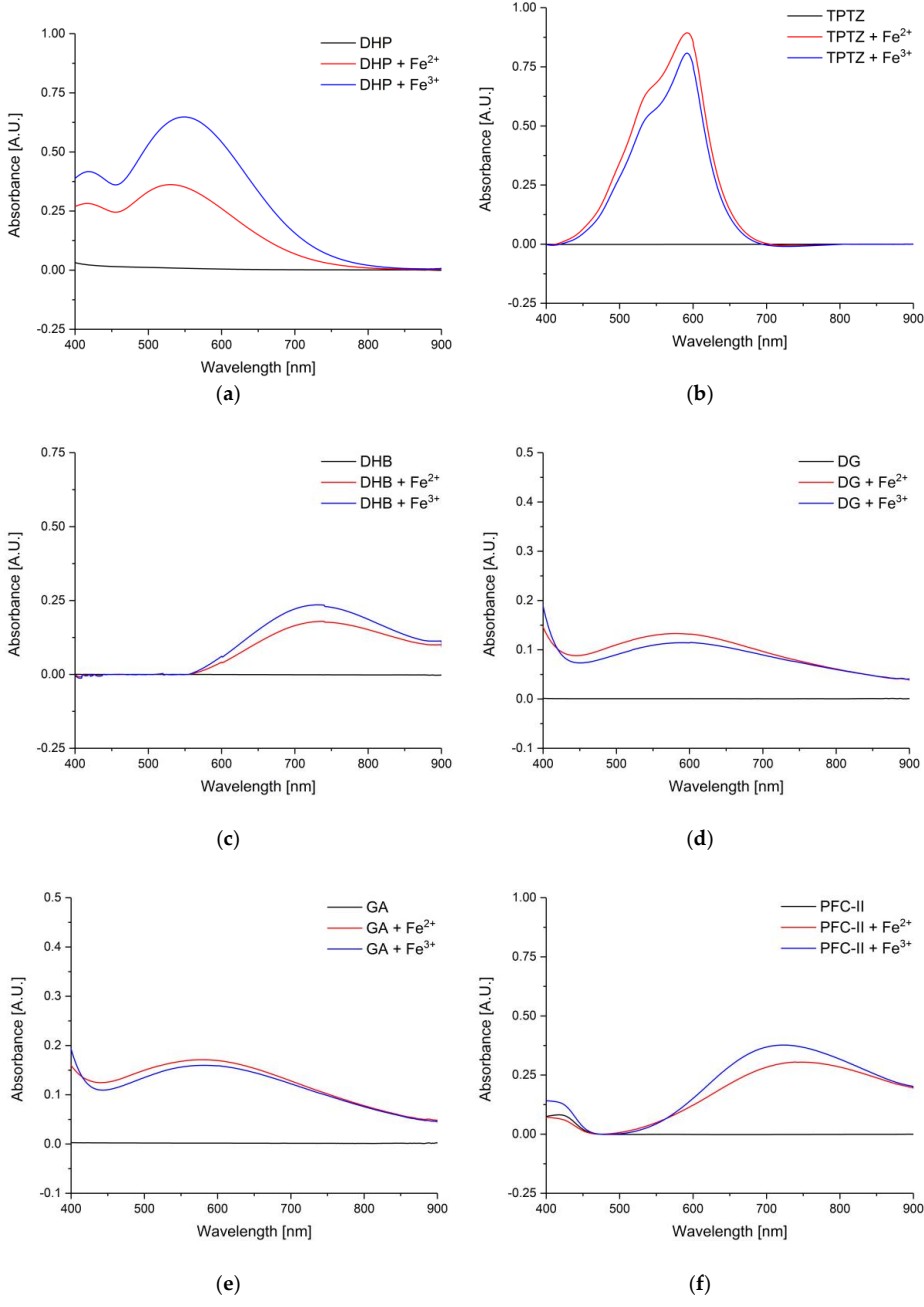

**Figure 7.** *Cont.*

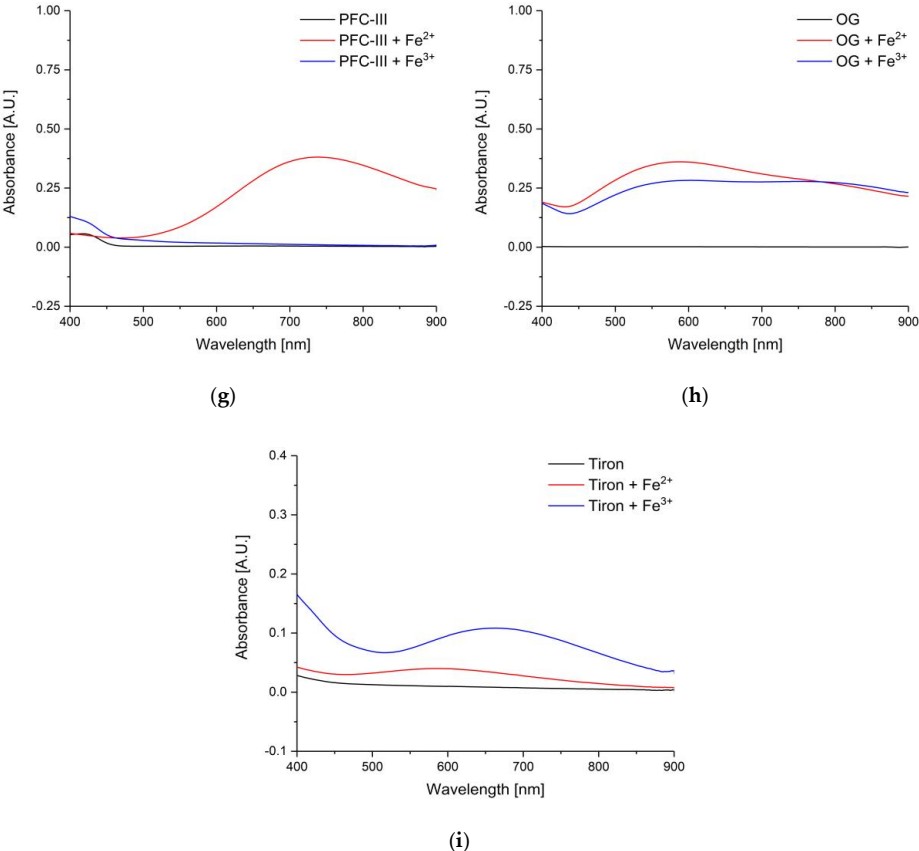

**Figure 7.** Black: UV–VIS spectra of the respective dyes, red: after the addition of 10 µL of a 10.3 mg/mL iron(II) chloride solution in diethyl ether, blue: after adding 10 µL iron(III) chloride solution in diethyl ether (10.3 mg/mL) to the gel/dye mixture. (**a**) DHP, (**b**) TPTZ, (**c**) DHB, (**d**) DG, (**e**) GA, (**f**) PFC-II, (**g**) PFC-III, (**h**) OG, and (**i**) Tiron.

The UV–VIS spectra are intended to provide information about which wavelength is best suited for the respective dye. DHP, TPTZ, DG, and GA show increased absorbance values at around 550 nm to 600 nm after reaction with iron ions, whereas DHB, PFC-II, and Tiron show best performance at around 700 nm. OG shows a similar increase over a wide spectral range.

However, conventional UV–VIS spectroscopy needs high amounts of dyes in order to observe a change in absorbance. Furthermore, a sensor for direct detection on-site is envisaged since corrosion processes may occur in inner parts, and can, therefore, not be observed via visible inspection.

*3.2. Sapphire Fiber-Optic Evanescent Wave Sensor*

3.2.1. Allan Deviation Analysis

Using the Allan–Werle deviation analysis as introduced by Allan in 1966 [29] and Werle et al. in 1993 [30], the $\delta$ value precision and, therefore, long-term drift influences, were evaluated. The laser was operated at 705 nm with a sampling interval of 28 measurements per minute with an entire measurement time of 4 h. In Figure 8, the time-dependent signal obtained using the sapphire fiber-optic sensor and the Allan deviation plot are shown. The Allan deviation is the square root of the Allan Variance [31]. With a 1 s measurement time, the Allan deviation revealed a precision of 0.72% since at short integration times the precision is influenced by white noise. By increasing the integration time, the precision can be enhanced up to 0.13% with an averaging time of 95 s. The maximum limit for the precision achieved for long-term measurements is described by $\tau_{max}$ with an averaging time of 200 s with 0.081%. Up to the optimum integration time, signal averaging leads to a decrease in standard

deviation and an improvement of the sensor resolution due to signal stability, therefore, the sensitivity of the sensor can be enhanced [32]. Further increasing average times are affected by drifts, which occur due to variation in the emission spectrum of the LED [31].

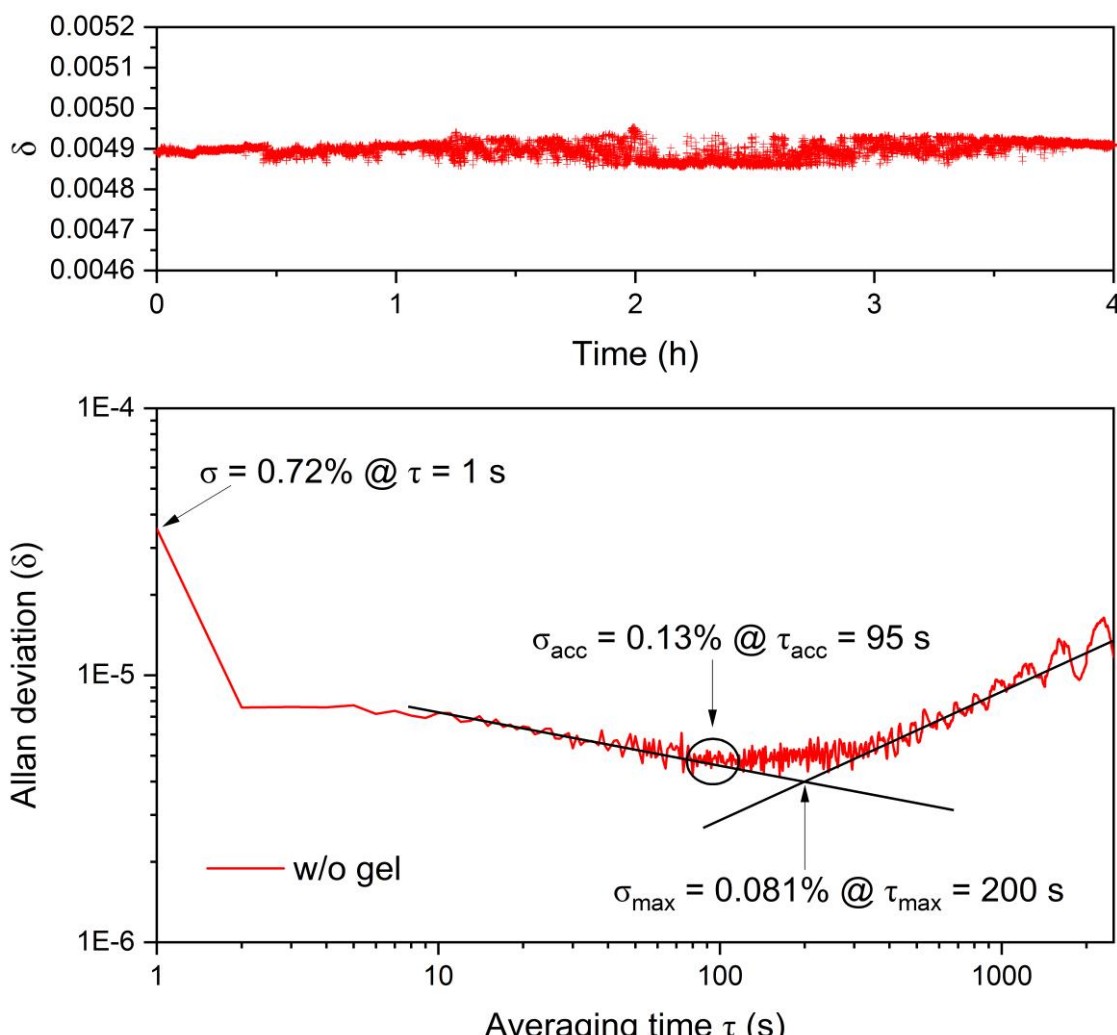

**Figure 8.** Allan variance analysis as a function of averaging time of the sapphire fiber-optic sensor for a 4 h time series at a wavelength of 705 nm. 1 s averaging results in a precision of 0.72%, the precision, i.e., limit for signal averaging is 0.13% with an averaging time of 95 s. The final limit for the precision is given by $\tau_{max}$ with 200 s.

### 3.2.2. Corrosion Detection

After the measuring cell was prepared, i.e., implementation of the sapphire fiber and addition of the gel/dye mixture, the corroded iron bar was added. The emerging iron ions start to react with the dye. In Figure 9, the time-dependent absorbance signal for TPTZ at laser diode wavelength of 520 nm is shown over a total period of 50 h. Within the first 6 h, and the last 10 h, one measurement per hour was performed. In between, measurements were executed every 5 h. The red line represents the applied logistic fit according to the following formula [33]:

$$y = \frac{S}{1 + a \cdot \exp(-kt)} \tag{12}$$

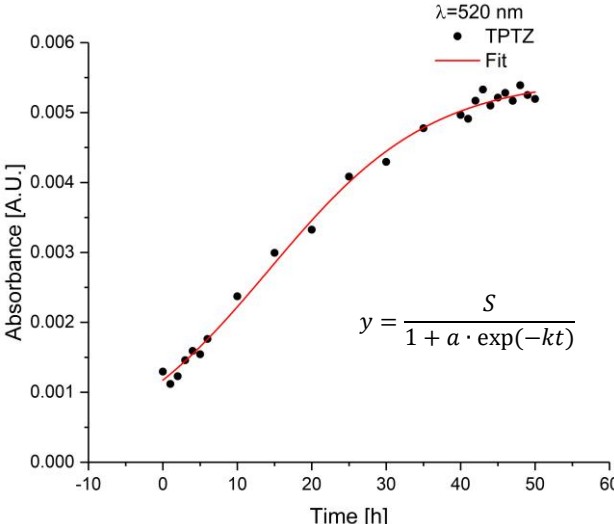

**Figure 9.** Absorbance values of sapphire fiber sensor as a function of time when exposed to the corroded iron bar at a wavelength of 520 nm. At the beginning, the bar was implemented inducing iron ions and, therefore, the start of reactions with the dye, i.e., TPTZ. After approx. 45 h, the signal stays constant.

With the maximum value $S$, the growth constant $k$, the time $t$ in hours, and a constant $a$, which can be calculated by $\frac{S}{b} - 1$, with the initial value $b$ at $t = 0$ h. The obtained values are summarized in Table 3. Within the first 3 h, no significant change in absorbance values were observed. After 4 h, the signal increased. Around 45 h, the signal reaches an equilibrium. Therefore, no reaction of iron ions with dye molecules occurred since no more iron ions are present or all dye molecules have already reacted.

**Table 3.** Parameters of logistic fit function for the time dependent absorbance signal of TPTZ exposed to a corroded iron bar measured at a wavelength of 520 nm.

|  | $S$ | $a$ | $k$ | $r^2$ |
|---|---|---|---|---|
| Logistic fit function | 0.00549 | 3.67986 | 0.09164 | 0.99584 |

Conventionally, within permanently installed systems, e.g., in bridges or reinforced concrete, corrosion does not occur frequently. Hence, for future application directly in the field, the proposed corrosion sensor is intended for long-term monitoring. If corrosion processes are induced, and reaction of iron ions with dye molecules from the sensor occur, the entire component and the measuring cell containing the gel/dye solution has to be replaced. The replacement of the measuring cell is very simple and cost-saving since only the fiber connectors have to be disconnected.

The nine dyes were tested for corrosion detection, i.e., iron ions, using the developed sensor based on sapphire fiber evanescent field spectroscopy. Therefore, all dyes were measured at three different wavelengths, i.e., 520 nm, 635 nm, and 705 nm as shown in Figure 10a–c, respectively. The obtained absorbance values of the gel/dye mixture (black), corresponding to $t = 0$ h, are compared to the absorbance values obtained after 50 h (red) after the corroded iron bar was implemented in the measuring cell. Obviously, after 50 h different responses to the produced iron ions were obtained. The higher the absorbance value after 50 h in comparison to the gel/dye mixture, i.e., the beginning of the measurement, the higher the sensitivity to iron ions. Best performance was obtained with TPTZ and OG at 520 nm and 705 nm, respectively. At 635 nm, DG and GA show a significantly increased absorbance value. In general, the results agreed with the UV–VIS spectra, e.g., the UV–VIS spectrum of DHP flattened towards 800 nm. Therefore, the absorbance values are almost similar at 705 nm, but at 635 nm a significant difference was observed. Therefore, depending on the availability of the dye and its costs and toxicity, TPTZ, DG, GA, and OG are most suitable dyes for detecting iron ions, and thus, the corrosion process.

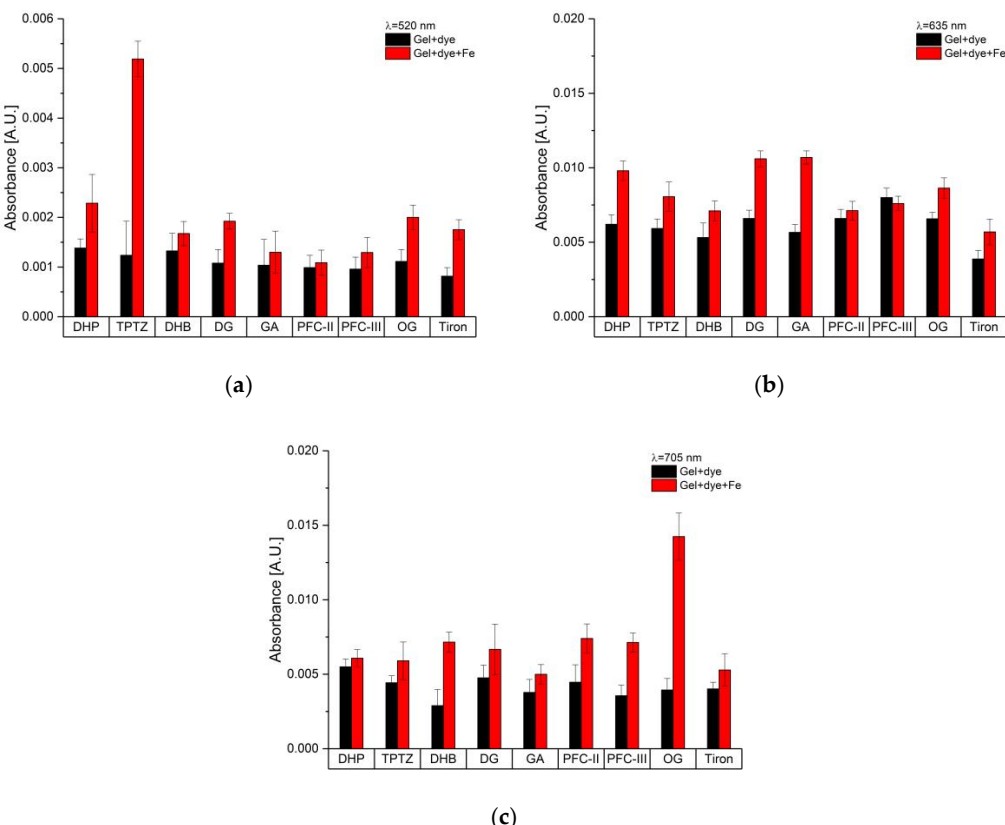

**Figure 10.** Comparison of absorbance values of gel/dye mixtures, i.e., at the beginning of the measurement (black) and after 50 h after the corroded iron bar was added (red) leading to a release of iron ions at (**a**) 520 nm, (**b**) 635 nm, and (**c**) 705 nm.

The repeatability of the developed visible-ATR sensor was tested using TPTZ as test dye due to its superior response at a wavelength of 520 nm. The gel/dye solution was prepared successively five times and a corroded iron bar added at the beginning of the measurement. The absorbance values after 50 h of measurement, i.e., exposure of the gel/dye mixture to the corroded iron bar are shown in Figure 11.

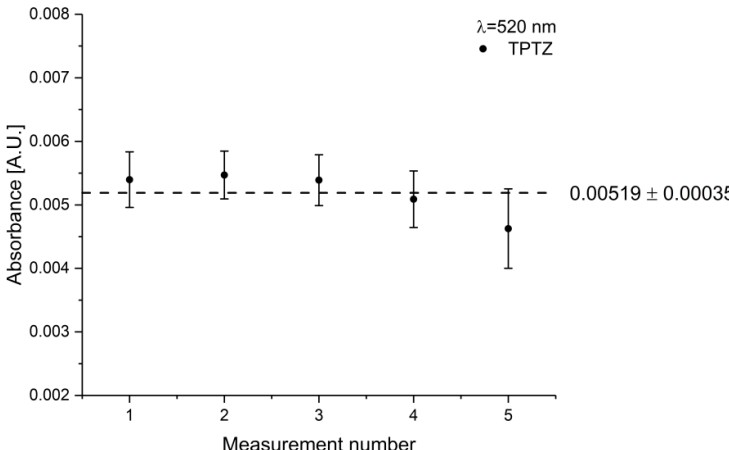

**Figure 11.** Repeatability of the sapphire fiber-based ATR sensor. The gel/dye mixture was prepared independently, and a corroded iron bar was added with the beginning of the measurement time. The data points at each measurement number represent the absorbance values of the gel/dye mixture exposed to the iron bar for 50 h.

### 3.2.3. Calibration

In order to establish a calibration function, different amounts of iron ions were added manually to the gel/dye mixture. As dye, OG was selected due to its superior behavior at 705 nm compared to the other investigated dyes. The amount of OG was 200 mg in 10 g gel. Therefore, a solution of 10 μL, 20 μL, 30 μL, 40 μL, and 50 μL of 0.01 mg iron(III) chloride in diethyl ether was added corresponding to an iron amount ranging from 0.01 to 0.05 mg. Each iron content was measured five times represented by the mean value and the standard deviation. The resulting calibration function is illustrated in Figure 12. The method revealed a goodness of fit ($r^2$-value) of 0.99694. The limit of detection (LOD) and limit of quantification (LOQ) was calculated according to IUPAC using three-times and 10-times the standard deviation of the blank, respectively [34]. The calculated LOD was 0.00386 mg and the calculated LOQ was 0.0001285 mg. Therefore, the developed sensor system shows a similar sensitivity to iron ions compared to UV–VIS spectroscopy using the same concentration range of iron ions and the same dye concentration (LOD is 0.00209 mg and LOQ was calculated to be 0.00693 mg). The results including the figures-of-merit are summarized in Table 4. To the best of our knowledge, this is the first time of detecting corrosion processes inducing the release of iron ions via sapphire fiber-optic sensors in the visible spectral region. Compared to a previous study operating in the mid-infrared in combination with diamond-like carbon-coated silicon wafers [11], the sensitivity was significantly increased from 13 μg/μL to 0.33 μg/mL.

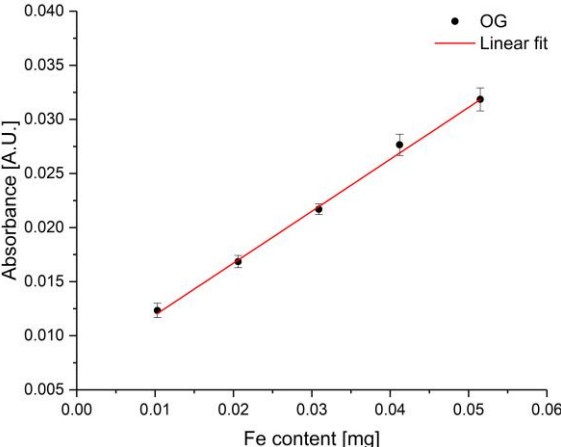

**Figure 12.** Established calibration function with the sapphire fiber-based corrosion sensor at five different amounts of iron ranging from 0.01 mg to 0.05 mg at a wavelength of 705 nm. Data points were obtained from five independent repetitive measurements.

**Table 4.** Summary of established calibration function obtained with the developed corrosion sensor compared to conventional UV–VIS spectroscopy with $r^2$-value, LOD and LOQ.

| Figure-of-Merit | FEWS Sensor | UV–VIS Spectroscopy |
|---|---|---|
| Calibration function | 0.48006x + 0.00711 | 13.17952x + 0.0282 |
| $r^2$ | 0.99694 | 0.99952 |
| LOD (mg) | 0.00386 | 0.00209 |
| LOQ (mg) | 0.01285 | 0.00693 |

## 4. Conclusions

A sensor operating in the visible spectral range for detection of iron ions released due to formation of rust was presented. The developed sensor is comprising of a laser diode as light source, an unclad sapphire fiber waveguide located inside a 3D printed measuring cell, and a photodiode for signal detection. Light emanating from the laser diode was guided via cladded silica fibers to the fiber-optical waveguide and to the detector. An iron bar was placed inside the measuring cell near the fiber-optical

waveguide surrounded by a gel which is working as a protecting agent, i.e., an inhibitor against corrosion. The gel was mixed with a dye, hence, if corrosion occurs and iron ions are released a complexation reaction takes place. Nine different dyes were investigated within this study. Since the sapphire fiber was placed next to the corroding material, i.e., corroded iron bar, corrosion can be detected early preventing, e.g., safety risks, due to a fast response. The dyes TPTZ, DG, GA, and OG showed the best behavior and a good sensitivity to iron ions released from the corroded iron bar. After about 4 h a significant increase in absorbance signal was observed. In particular, OG provides the best sensitivity at a wavelength of 705 nm. The developed sensor provides detection limits comparable to those of conventional UV–VIS spectroscopy in the range of 3.86 µg and a quantification limit of 12.85 µg when using OG as dye. OG showed the highest sensitivity to iron ions compared to the other investigated dyes. The costs of the dye are not high since only 200 mg per 10 g gel were used. Therefore, real-world application as a mechanically robust corrosion detection system on site is enabled by placing the FEWS sensor next to corrosion susceptible materials or components or wherever corrosion may occur.

**Author Contributions:** Conceptualization: D.T., A.K., and B.M.; methodology: D.T.; data curation: D.T.; writing—original draft preparation: D.T.; writing—review and editing: A.K. and B.M.; supervision: B.M.; funding acquisition: B.M. All authors have read and agreed to the published version of the manuscript.

**Funding:** This research was funded by Wertec-GmbH.

**Acknowledgments:** The authors gratefully acknowledge support by Wertec-GmbH providing the gel used for protecting components from corrosion.

**Conflicts of Interest:** The authors declare no conflict of interest.

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
