# Peer review of "Monitoring Corrosion Processes via Visible Fiber-Optic Evanescent Wave Sensor"

_chemosensors, doi:10.3390/chemosensors8030076_

Round 1
Reviewer 1 Report
The paper from Turkmen et al. reports an experimental study of using unclad sapphire fiber as evanescent wave sensor for corrosion process especially ferrous ions. The paper is overall clear. Some concerns that are listed below should be addressed.
- How is the repeatability of the proposed fiber sensor in this study? And how is reversibility of the sensor? Seems like after the Fe ions reacts with the gel+dye solution, the gel+dye solution has to be replaced for another new sensing measurements. The authors should comment on these. And how this proposed sensor could be used on-site in real application if you have to replace the gel+dye solution frequently.
- How is the sapphire fiber couple with silica fiber, simply physical butting using some kind of connector, or just free space, or fuse splicing? The authors should provide more details on that, including silica fiber core/cladding size, fiber facet optical coupling loss, etc. And how is the optical coupling repeatability from different fiber sensors? The author should comment on how they normalize/calibrate the background ATR power to make sure fiber sensors are comparable to each other.
- What is the quantitative sensitivity for the proposed fiber sensor in this study? The authors should estimate this, and compare with similar study, so the significance of the contribution can be more clear.
Author Response
Dear reviewer,
thank you for your thorough review of the manuscript and your important comments and suggestions. Please find our response attached.
Kind regards.

Reviewer 2 Report
Please see the attached report.

Author Response
Dear reviewer,
thank you for your thorough review of the manuscript and your important comments and suggestions. You can find an explanation point-by-point below.
Kind regards.

Round 2
Reviewer 1 Report
The reviewer is satisfied with the revision. Accept as it is.
Author Response
Dear Sir or Madam,
thank you very much for reviewing the manuscript.
Kind regards
Reviewer 2 Report
The revised version of the manuscript entitled “Monitoring Corrosion Processes via Visible Fiber-optic Evanescent Wave Sensor” has been surely improved. However, there is still the issue regarding the LOD and LOQ which deserves a further modification given their critical relevance in the assessment of sensing performance:
- (lines 341-342) Actually, the definition of LOD and LOQ provided by the Authors is not totally correct and could underpin misleading information for the readers. According to ref. “P. Zubiate et al., Biosensors and Bioelectronics X, vol. 2, p. 100026, 2019,” and both the ACS and IUPAC recommendations, the correct definitions are the following:
- LOD: the signal of the blank plus three times (why 3.3??) the standard deviation of the blank. In a more conservative case, one can consider the mean standard deviation or, at least, the maximum standard deviation.
- LOQ: the signal of the blank plus ten times the standard deviation of the blank. Again, in a more conservative case, one can consider the mean standard deviation or, at least, the maximum standard deviation.
After this further improvement in the revised manuscript, I think the manuscript can be accepted for publication in Chemosensors.
Author Response
Dear Sir or Madam,
The definitions for LOD and LOQ were revised.
Kind regards.